# Synthesis of Daidzein Glycosides, α-Tocopherol Glycosides, Hesperetin Glycosides by Bioconversion and Their Potential for Anti-Allergic Functional-Foods and Cosmetics

**DOI:** 10.3390/molecules24162975

**Published:** 2019-08-16

**Authors:** Yuya Fujitaka, Hiroki Hamada, Daisuke Uesugi, Atsuhito Kuboki, Kei Shimoda, Takafumi Iwaki, Yuya Kiriake, Tomohiro Saikawa

**Affiliations:** 1Department of Life Science, Faculty of Science, Okayama University of Science, 1-1 Ridai-cho, Kita-ku, Okayama 700-0005, Japan; 2Department of Biochemistry, Faculty of Science, Okayama University of Science, 1-1 Ridai-cho, Kita-ku, Okayama 700-0005, Japan; 3Department of Biomedical Chemistry, Faculty of Medicine, Oita University, 1-1 Hasama-machi, Oita 879-5593, Japan; 4Department of Biophysics, Faculty of Medicine, Oita University, 1-1 Hasama-machi, Oita 879-5593, Japan; 5Faculty of Medicine and Health Sciences, Yamaguchi University, 1-1-1 Minamikogushi, Ube-shi, Yamaguchi 755-8505, Japan; 6Department of Nursing, Junshin Gakuen University, 1-1-1 Tikushigaoka, Fukuoka-shi, Minami-ku, Fukuoka 815-8510, Japan

**Keywords:** daidzein, α-tocopherol, hesperetin, β-glycoside, anti-allergic activity, tyrosinase inhibitory activity

## Abstract

Daidzein is a common isoflavone, having multiple biological effects such as anti-inflammation, anti-allergy, and anti-aging. α-Tocopherol is the tocopherol isoform with the highest vitamin E activity including anti-allergic activity and anti-cancer activity. Hesperetin is a flavone, which shows potent anti-inflammatory effects. These compounds have shortcomings, i.e., water-insolubility and poor absorption after oral administration. The glycosylation of bioactive compounds can enhance their water-solubility, physicochemical stability, intestinal absorption, and biological half-life, and improve their bio- and pharmacological properties. They were transformed by cultured *Nicotiana tabacum* cells to 7-β-glucoside and 7-β-gentiobioside of daidzein, and 3′- and 7-β-glucosides, 3′,7-β-diglucoside, and 7-β-gentiobioside of hesperetin. Daidzein and α-tocopherol were glycosylated by galactosylation with β-glucosidase to give 4′- and 7-β-galactosides of daidzein, which were new compounds, and α-tocopherol 6-β-galactoside. These nine glycosides showed higher anti-allergic activity, i.e., inhibitory activity toward histamine release from rat peritoneal mast cells, than their respective aglycones. In addition, these glycosides showed higher tyrosinase inhibitory activity than the corresponding aglycones. Glycosylation of daidzein, α-tocopherol, and hesperetin greatly improved their biological activities.

## 1. Introduction

Daidzein is the most widely studied isoflavone and is found in beans such as soybean, sweet red bean, and kidney bean. It exhibits anti-oxidative, anti-inflammatory, and anti-aging action and has chemopreventive effects in several biological systems, such as cancer prevention [1,2,3,4,5,6,7,8,9,10]. α-Tocopherol is found in wheat germ. It has chemopreventive activities against cancer such as breast and prostate cancers [11,12,13,14,15]. Hesperetin, which occurs in citrus fruits and flowers, is a bioactive flavonoid (vitamin P) that has been well documented for its medicinal properties as an important Chinese traditional medicine [16]. Available clinical information on hesperetin includes its effects on the blood–brain barrier, signal transduction pathway, and certain kinds of cancer [17,18,19,20,21]. Hesperetin has been used in cosmetics as it has antioxidant and anti-allergic activities. Despite the bio- and physiological activities of these compounds, their use as medicines, functional-foods, and cosmetics has been limited due to their insolubility in water and poor absorption after oral administration.

Several flavonoids have been reported to possess a histamine release-inhibitory activity on rat peritoneal mast cells [22]. However, comparative studies on the anti-allergic activity of many kinds of flavonoid glycosides have not yet been done. The release of inflammatory mediators, such as histamine, from mast cell or basophils, is mediated by cell-surface receptors for immunoglobulin E (IgE), and stimulation by antigen results in allergic reactions called immediate-type hypersensitivity. In vitro anti-allergic screening has generally been performed by using rat peritoneal mast cells [22]. On the other hand, melanogenesis is a physiological process resulting in the production of melanin pigment, which plays an important role in the prevention of sun-induced skin injury. Although the melanin production in human skin is a major defense mechanism against UV light, the accumulation of an excess of epidermal pigmentation can cause various hyperpigmentation disorders, such as melasma, age spots, and sites of actinic damage. Tyrosinase (EC 1.14.18.1) is a copper-containing enzyme widely distributed in nature. It catalyzes two distinct reactions involving molecular oxygen in the melanin synthesis, the hydroxylation of l-tyrosine to l-dopa and the oxidation of l-dopa to dopaquinone. This dopaquinone is highly reactive and can polymerize spontaneously to form melanin in a series of reaction pathways. Accordingly, the regulation of melanin synthesis by the inhibition of tyrosinase to prevent hyperpigmentation has been a recent subject of many studies [23].

Plant cell cultures are ideal systems for propagating rare plants and for studying the biosynthesis of secondary metabolites such as flavors, pigments, and agrochemicals, except for a very limited number of compounds (e.g., pyrethrins, bialaphos, and nicotin). Furthermore, the bioconversion of various organic compounds has been investigated as a target in the biotechnological application of plant cell culture systems. Plant cultured cells can be used to convert organic molecules to more useful compounds by catalyzing hydrolysis, oxidation, reduction, esterification, isomerization, and glycosylation reactions. A recent paper reported that alkyl esters of caffeic acid, especially propyl caffeate, have stronger antioxidant activity than caffeic acid [24]. On the other hand, the glycosylation of bioactive compounds can enhance their water-solubility, physicochemical stability, intestinal absorption, biological half-life, and improve their bio- and pharmacological properties [25,26,27,28,29,30,31,32,33].

We report here the syntheses of glycosides of daidzein, α-tocopherol, and hesperetin (Figure 1) by bioconversion with cultured plant cells and β-glucosidase. In addition, we report the physiological properties of glycosides of daidzein, α-tocopherol, and hesperetin such as their histamine-release inhibitory activity (anti-allergic activity) and tyrosinase inhibitory activity.

## 2. Results

### 2.1. Synthesis of Daidzein Glycosides and Hesperetin Glycosides

Daidzein (**1**) was subjected to a bioconversion system using cultured cells of *N. tabacum*. Two products, **2** and **3**, were isolated from the cells by extraction with MeOH after two days of incubation. Based on the analyses of mass and nuclear magnetic resonance (NMR) analyses, the products were determined to be daidzein 7-β-glucoside (**2**) and daidzein 7-β-gentiobioside (**3**), which were known compounds (Figure 2) [34].

On the other hand, only a small amount of the glycosylation products in the MeOH extracts of *N. tabacum* cells treated with α-tocopherol (**6**) was obtained. It was not enough for analyses of biological activities.

Furthermore, biotransformation of hesperetin (**8**) with *N. tabacum* cells was carried out. Products **9**–**12** were isolated from the MeOH extracts from cultured *N. tabacum* cells. The structure of compounds **9**–**12** was identified as hesperetin 3′-β-glucoside (**9**), hesperetin 7-β-glucoside (**10**), hesperetin 3′,7-β-diglucoside (**11**), and hesperetin 7-β-gentiobioside (**12**), which have been reported before [35], (Figure 2), based on analyses of mass and nuclear magnetic resonance (NMR) spectra.

### 2.2. Enzymatic Galactosylation with β-Glucosidase Isolated from Sweet Almond

#### 2.2.1. Synthesis of Daidzein Galactosides and α-Tocopherol Galactoside

Galactosylation of daidzein (**1**) was performed by enzymatic synthetic procedure. β-Glucosidase isolated from sweet almond, which can catalyze glycosylation, i.e., reverse hydrolysis, such as glucosylation and galactosylation [36], was used for the synthesis of daidzein galactosides and the conditions were optimized at pH 7, over 72 h, 110 AU of enzyme. Based on the analyses of HRESIMS, ^1^H- and ^13^C-NMR, and HMBC spectra, the products were determined to be daidzein 4′-β-galactoside (**4**) and daidzein 7-β-galactoside (**5**) (Figure 3). Products **4** and **5** were two new compounds. No glucosylation of the substrates occurred, because d-galactose was used as the galactose donor.

On the other hand, α-tocopherol (**6**) was subjected to the same galactosylation procedures as described above. Product **7** was isolated by preparative HPLC. The chemical structure of the product **7** was identified as α-tocopherol 6-β-galactoside (Figure 3) [36], based on the analyses of ESIMS, ^1^H- and ^13^C-NMR, and HMBC spectra.

No galactosylation products were isolated from the reaction mixture of enzymatic transformation of hesperetin (**8**). It might be explained by the substrate specificity of sweet almond β-glucosidase.

#### 2.2.2. Determination of Chemical Structure of New Daidzein Galactosides

The HRESIMS spectrum of **4** showed a pseudomolecular ion [M + Na]^+^ peak at *m*/*z* (439.265), consistent with a molecular formula of C_21_H_20_O_9_ (calcd. 439.368 for C_21_H_20_O_9_Na). The ^1^H-NMR spectrum of **4** had a signal at *δ* 4.91 (1H, *d*, *J* = 8.0 Hz) corresponding to its attachment to the anomeric carbon (C-1″) (Appendix A). The ^13^C-NMR spectrum of **4** exhibited the anomeric carbon signal at *δ* 103.0 (Appendix A). This ^13^C chemical shift of the anomeric carbon at *δ* 103.0 indicates the presence of *O*-galactoside in the structure of **4** [36]. From the coupling pattern of the proton signals and the chemical shifts of the carbon resonances due to the sugar moiety, the sugar component in **4** was determined to be β-d-galactopyranose. Hydrolysis of **4** using β-galactosidase gave daidzein (**1**) as the product. This finding shows that the product has *O*-β-galactosylation moiety. The HMBC correlation (Figure 4) was observed between the anomeric proton signal at *δ* 4.91 (H-1″) and the carbon signal at *δ* 159.1 (C-4′) to establish that the galactopyranosyl residue was attached to the 4′-hydroxy group of **1** (Appendix A). Thus, the structure of **4** was determined to be daidzein 4′-β-galactoside.

The HRESIMS spectra of **5** included a pseudomolecular ion [M + Na]^+^ peak at *m*/*z* (439.248), indicating that the product consisted of one substrate and one hexose. The sugar component in the product was determined to be galactose on the basis of the chemical shifts of their carbon signals (Appendix A) [36]. The ^1^H-NMR spectrum showed a proton signal at *δ* 5.05 (1H, *d*, *J* = 7.2 Hz), indicating that the galactoside linkage in the compound had β-orientation (Appendix A). The HMBC spectra of **5** included the correlation between the proton signal at *δ* 5.05 (H-1″) and the carbon signal at *δ* 163.6 (C-7) (Appendix A). These data indicate that **5** was β-galactosyl analogue of **1**, the sugar moiety of which was attached to the 7-position of **1** (Figure 4). Thus, the structure of **5** was determined to be daidzein 7-β-galactoside.

### 2.3. Biological Effects of Daidzein Glycosides, α-Tocopherol Glycoside, and Hesperetin Glycosides

#### 2.3.1. Suppression for Histamine Release from Rat Peritoneal Mast Cells

The anti-allergic activities of daidzein and its glycosides were examined to clarify the effects of glycosylation of daidzein on its physiological activities. The inhibitory activities of daidzein (**1**), daidzein 7-β-glucoside (**2**), daidzein 7-β-gentiobioside (**3**), daidzein 4′-β-galactoside (**4**), and daidzein 7-β-galactoside (**5**) for compound 48/80-induced histamine release from rat peritoneal mast cells were investigated (Table 1). Compound 48/80-induced histamine release from rat peritoneal mast cells was inhibited by daidzein (**1**) with a %inhibition of 58%. On the other hand, the inhibitory activity of daidzein 7-β-glucoside (**2**) toward histamine release from rat peritoneal mast cells was higher than that of the aglycone daidzein (**1**). The inhibitory activity of daidzein 7-β-gentiobioside (**3**) was higher than that of the daidzein glucoside **2**. Additionally, daidzein 4′-β-galactoside (**4**) and daidzein 7-β-galactoside (**5**) showed higher inhibitory activities toward histamine release than daidzein gentiobioside **3**.

In addition, α-tocopherol 6-β-galactoside (**7**) had stronger inhibitory activity toward histamine release than α-tocopherol (**6**).

The %inhibition of hesperetin 3′-β-glucoside (**9**) and hesperetin 7-β-glucoside (**10**) was higher than that of hesperetin (**8**). Furthermore, hesperetin 3′,7-β-diglucoside (**11**) showed higher inhibitory activity toward histamine release than hesperetin glucosides **9** and **10**. The inhibitory activity toward histamine release of hesperetin 7-β-gentiobioside (**12**) was stronger than hesperetin diglucoside **11**. The anti-allergic activities were examined at pH 7.4 and 37 °C according to the reported methods [37]. The pH and temperature of the reaction mixture may have an effect on anti-allergic activities. Effects of pH and temperature on anti-allergic activities of the obtained glycosides should be investigated and reported on in the near future.

#### 2.3.2. Tyrosinase Inhibitory Activity

The IC_50_ value of daidzein (**1**) for tyrosinase inhibitory activity was 392 μM (Table 2). The glucoside of daidzein, daidzein 7-β-glucoside (**2**), showed high inhibitory activity against tyrosinase. This result indicates that the glucosylation of daidzein improved its tyrosinase inhibitory activity. In addition, the glucosylation of daidzein glucoside **2** to daidzein 7-β-gentiobioside (**3**) enhanced its tyrosinase inhibitory activity. Overall, daidzein 7-β-galactoside (**5**) showed the highest inhibitory activity against tyrosinase among daidzein compounds tested.

A similar tendency was found in the case of α-tocopherol and its glycoside. The tyrosinase inhibitory activity of α-tocopherol 6-β-galactoside (**7**) was higher than that of α-tocopherol (**6**).

Additionally, tyrosinase inhibition by hesperetin 3′-β-glucoside (**9**) and hesperetin 7-β-glucoside (**10**) was stronger than that by hesperetin (**8**). Hesperetin 3′,7-β-diglucoside (**11**) had stronger tyrosinase inhibitory activity than hesperetin glucosides **9** and **10**. Particularly, hesperetin 7-β-gentiobioside (**12**) showed the strongest tyrosinase inhibition among the hesperetin compounds tested. Effects of pH and temperature on tyrosinase inhibitory activity of the obtained glycosides should be investigated and reported in the near future.

## 3. Discussion

Thus, the glycosylation derivatives of daidzein, α-tocopherol, and hesperetin were prepared by bioconversion with cultured *N. tabacum* and β-glucosidase. Cultured *N. tabacum* cells glycosylated daidzein and hesperetin to 7-β-glucoside and 7-β-gentiobioside of daidzein, and 3′- and 7-β-glucosides, 3′,7-β-diglucoside, and 7-β-gentiobioside of hesperetin. A recent paper reported that cultured *E. perriniana* cells converted daidzein into 7-β-glucoside and 7-β-gentiobioside of daidzein [34]. The bioconversion system of *N. tabacum* cells is the same as *E. perriniana* cells. It was reported that *E. perriniana* cells glycosylated hesperetin to hesperetin 3′-*O*-β-glucoside, hesperetin 3′,7-*O*-β-diglucoside, hesperetin 7-*O*-β-gentiobioside, hesperetin 5-*O*-β-glucoside, hesperetin 7-*O*-β-glucoside, and hesperetin 7-*O*-β-rutinoside [35]. The bioconversion pathway of hesperetin by cultured *N. tabacum* cells is quite different from that of cultured *E. perriniana* cells. Daidzein and α-tocopherol were galactosylated by β-glucosidase to give 4′- and 7-β-galactosides of daidzein, which were new compounds, and α-tocopherol 6-β-galactoside.

Recently, biocatalytic synthesis of daidzein glycoside has been reported. The cell culture of *Staphylococcus saprophyticus* CQ16 glycosylated daidzein to daidzein 7-*O*-β-glucoside [38]. In addition, the biocatalytic glucosylation of 8-hydroxydaidzein to its 7-*O*-β-glucoside and 8-*O*-β-glucoside has been reported. 8-Hydroxydaidzein has been proven to possess some important bioactivities, however, the low aqueous solubility and stability of 8-hydroxydaidzein limit its pharmaceutical and cosmeceutical applications. The glucosylation of 8-hydroxydaidzein by glucosyltransferase from *Bacillus subtilis* ATCC 6633 improved its drawbacks in solubility and stability [39]. These microbiological glycosylation of isoflavones gave only mono glucoside as the product. Compared with microbiological glycosylation, bioconversion of isoflavone by plant cells is a convenient method to prepare diverse glycosides, such as mono- and di-glycosides including gentoibioside.

Recently, it was reported that anti-allergic activity of stilbene compounds, such as resveratrol, pterostilbene, and piceatannol, were enhanced by glycoside modification [40,41]. An analysis of the inhibitory activities of daidzein, daidzein glycosides, α-tocopherol, α-tocopherol glycoside, hesperetin, and hesperetin glycosides toward histamine release from rat peritoneal mast cells showed that glycosylation of the isoflavone daidzein, vitamin E (α-tocopherol), and flavone hesperetin improved their anti-allergic activities. α-Tocopherol 6-β-galactoside had the highest inhibitory activity toward histamine release among the compounds tested. Kaempferol was used as a positive control, the inhibitory activity toward histamine release of which was 80%. α-Tocopherol 6-β-galactoside showed higher anti-allergic activity than kaempferol. Improvement of anti-allergic activity was achieved by glycosylation for not only stilbene compounds but also daidzein, α-tocopherol, and hesperetin.

Uesugi and his co-workers recently reported that glycosylation of stilbene compounds, such as resveratrol, pterostilbene, and pinostilbene, increased their tyrosinase inhibitory activity [41]. The tyrosinase inhibitory activities of the daidzein glycosides, α-tocopherol glycoside, and hesperetin glycosides were higher than those of their respective aglycones. α-Tocopherol 6-β-galactoside showed the strongest tyrosinase inhibitory activity among the compounds tested. Kojic acid was used as a positive control, the tyrosinase inhibitory activity of which was IC_50_ = 35. Although the tyrosinase inhibitory activity of α-tocopherol 6-β-galactoside was slightly lower than that of kojic acid, α-tocopherol 6-β-galactoside acts as a strong tyrosinase inhibitor. The present results suggest that the galactosylated compounds might be useful as effective skin-whitening agents with strong tyrosinase inhibitory and anti-allergic activities. Tyrosinase inhibitory activity of not only stilbene compounds but also daidzein, α-tocopherol, and hesperetin, was enhanced by glycosylation. This is the first report that describes the anti-allergic activity of daidzein galactosides, which are new, and the inhibitory activity of α-tocopherol galactoside toward tyrosinase. The structure–activity relationship of the glycosides is now in progress in our laboratory.

## 4. Materials and Methods 

### 4.1. Analyses

The structures of products were determined based on the analysis of HRESIMS, ^1^H- and ^13^C-NMR, and HMBC spectra. The ^1^H- and ^13^C-NMR, and HMBC spectra were recorded using a JNM-ECS400 spectrometer (JEOL Ltd., Tokyo, Japan) in CD_3_OD solutions, and chemical shifts are expressed in δ (ppm) with reference to TMS. The HRESIMS spectra were measured using a JMS-700 MStation (JEOL) in CH_3_OH solution.

### 4.2. Glycosylation by Cultured Plant Cells

The substrate was transformed by using plant cultured cells of *N. tabacum* as biocatalysts. The cultured plant cells of *P. americana* were sub-cultured at 4-week intervals on solid medium containing 2% glucose, 1 ppm 2,4-dichlorophenoxyacetic acid, and 1% agar (adjusted to pH 5.7) in the dark. A suspension culture was started by transferring 20 g of the cultured cells to 300 mL of liquid MS medium in a 500 mL-conical flask. The cultured cells in the stationary growth phase have been used for experiments. To a 300 mL flask containing 100 mL of the culture medium and suspension cultured cells (25 g) was added 15 mg of substrate, i.e., daidzein, α-tocopherol, or hesperetin. The culture was incubated at 25 °C for 2 days on a rotary shaker (120 rpm). After the incubation period, the cells and medium were separated by filtration with suction. The filtered medium was extracted with ethyl acetate (AcOEt). The cells were extracted by homogenization with MeOH, and the resulting extract was concentrated. The residue was partitioned between H_2_O and AcOEt. The AcOEt layer was evaporated and the residue was re-dissolved in MeOH and purified by preparative high-performance liquid chromatography (HPLC) (column: CrestPak C18S; flow rate: 1.0 mL/min; column temperature: 40 °C).

### 4.3. Galactosylation by Enzyme

A typical galactosylation procedure was performed as follows. Syntheses of daidzein galactosides involved refluxing daidzein (**1**) (0.25 mmol) with 0.5 mmol d-galactose in 100 ml di-isopropyl ether in the presence of 25–210 AU β-glucosidase and 0.01 M pH 4–8 buffer for 72 h at 68 °C. After the reaction, the solvent was evaporated and the enzyme denatured at 100 °C by holding in a boiling water-bath for 10 min. The product glycoside was dissolved in water, extracted with ethylacetate, concentrated, and subjected to preparative HPLC to afford daidzein 4′-β-galactoside (**4**) and daidzein 7-β-galactoside (**5**), respectively.

The spectral data of products **4** and **5**, which were new compounds, are as follows.

Daidzein 4′-β-galactoside (**4**): HRESIMS [M + Na]^+^: *m*/*z* 439.265 (439.368 calcd. for C_21_H_20_O_9_Na); ^1^H-NMR (400 MHz, CD_3_OD): 3.58–3.92 (6H, multiplet, H-2″-H-6″), 4.91 (1H, doublet, *J* = 8.0 Hz, H-1″), 6.69 (1H, doublet, *J* = 2.4 Hz, H-8), 6.83 (1H, double-doublet, *J* = 8.4, 2.0 Hz, H-6), 7.16–7.18 (2H, multiplet, H-3′, H-5′), 7.46–7.48 (2H, multiplet, H-2′, 6′), 7.97 (1H, doublet, *J* = 8.8 Hz, H-5), 8.10 (1H, singlet, H-2); ^13^C-NMR (100 MHz, CD_3_OD): 62.4 (C-6″), 70.3 (C-4″), 72.3 (C-2″), 74.9 (C-3″), 77.0 (C-5″), 103.0 (C-1″), 103.8 (C-8), 116.2 (C-10), 117.7 (C-3′, C-5′), 119.0 (C-6), 125.3 (C-3), 127.7 (C-1′), 128.0 (C-5), 131.4 (C-2′, 6′), 154.5 (C-2), 159.1 (C-4′), 160.6 (C-9), 169.9 (C-7), 178.0 (C-4).

Daidzein 7-β-galactoside (**5**): HRESIMS [M + Na]^+^: *m*/*z* 439.248 (439.368 calcd. for C_21_H_20_O_9_Na); ^1^H-NMR (400 MHz, CD_3_OD): 3.62–3.94 (6H, multiplet, H-2″-H-6″), 5.05 (1H, doublet, *J* = 7.2 Hz, H-1″), 6.83–6.85 (2H, multiplet, H-3′, H-5′), 7.21 (1H, double-doublet, *J* = 8.8, 2.4 Hz, H-6), 7.25 (1H, doublet, *J* = 2.4 Hz, H-8), 7.36–7.38 (2H, multiplet, H-2′, 6′), 8.13 (1H, doublet, *J* = 8.8 Hz, H-5), 8.18 (1H, singlet, H-2); ^13^C-NMR (100 MHz, CD_3_OD): 62.5 (C-6″), 70.2 (C-4″), 72.1 (C-2″), 74.8 (C-3″), 77.3 (C-5″), 102.5 (C-1″), 105.0 (C-8), 116.5 (C-3′, C-5′), 117.1 (C-6), 120.2 (C-10), 123.8 (C-1′),126.3 (C-3), 128.3 (C-5), 131.4 (C-2′, 6′), 155.0 (C-2), 159.3 (C-4′, C-9), 163.6 (C-7), 178.1 (C-4).

### 4.4. Inhibition of Histamine Release from Rat Peritoneal Mast Cells

The effects of daidzein, α-tocopherol, hesperetin, and their glycosides on compound 48/80-induced histamine release from rat peritoneal mast cells were examined as follows. Peritoneal mast cells were collected from the abdominal cavities of rats (Male Wistar rats, Nippon SLC) and purified to a level higher than 95%. The purified mast cells were suspended in a physiological buffered solution (PBS) containing 145 mM NaCl, 2.7 mM KCl, 1.0 mM CaCl_2_, 5.6 mM glucose, and 20 mM 4-(2-hydroxyethyl)-1-piperazineethanesulfonic acid (HEPES) (pH 7.4) to give approximately 10^4^ mast cells/mL. Cell viability was always greater than 90% as judged by the trypan blue exclusion test. Mast cells were preincubated with the test compound (1 μM) for 15 min at 37 °C, and subsequently exposed to compound 48/80 at 0.35 μg/mL. Histamine release was determined by a fluorometric assay, and was expressed as a percentage of total histamine [37].

### 4.5. Tyrosinase Assay

Mushroom tyrosinase (EC 1.14.18.1) (Sigma-Aldrich Chemical Co., St. Louis, MO, USA) was used for the tyrosinase assay, with either l-DOPA or l-tyrosine as a substrate. In spectrophotometric experiments, enzyme activity was taken as the initial velocity (Vi) monitored by observing dopachrome formation at 475 nm with a UV spectrophotometer at 30 °C. All samples were dissolved in ethanol at 10 mM. First, 200 μL of a 2.7 mM l-tyrosine or 5.4 mM l-DOPA aqueous solution was mixed with 2687 μL of 0.25 M phosphate buffer (pH 6.8). Next, 100 μL of the sample solution and 13 μL of the same phosphate buffer solution containing mushroom tyrosinase (144 units) were added to the mixture. The inhibitor concentration that gave a 50% loss of activity (IC_50_) was obtained by fitting the experimental data to the logistic curve.

## 5. Conclusions

The glycosides of daidzein, α-tocopherol, and hesperetin were synthesized by bioconversion procedures. These two bioconversion systems prepared nine glycosides of these compounds, i.e., daidzein 7-β-glucoside, daidzein 7-β-gentiobioside, hesperetin 3′-β-glucoside, hesperetin 7-β-glucoside, hesperetin 3′,7-β-diglucoside, hesperetin 7-β-gentiobioside, daidzein 4′-β-galactoside, daidzein 7-β-galactoside, and α-tocopherol 6-β-galactoside. Daidzein 4′-β-galactoside and daidzein 7-β-galactoside were two new compounds. Glycosylation of daidzein, α-tocopherol, and hesperetin much improved their biological activities such as suppression activity toward histamine release from rat peritoneal mast cells and tyrosinase inhibitory activity. The diglucoside derivatives showed stronger activity than glucoside derivatives. The gentiobioside derivative had stronger activity than the diglucoside derivative. The galactoside derivative exerted the strongest physiological activity among the glycosides tested.

## Figures and Tables

**Figure 1 molecules-24-02975-f001:**
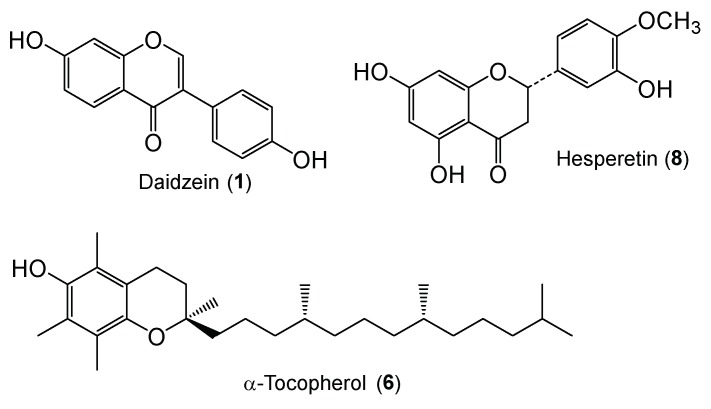
Chemical structures of daidzein, α-tocopherol, and hesperetin.

**Figure 2 molecules-24-02975-f002:**
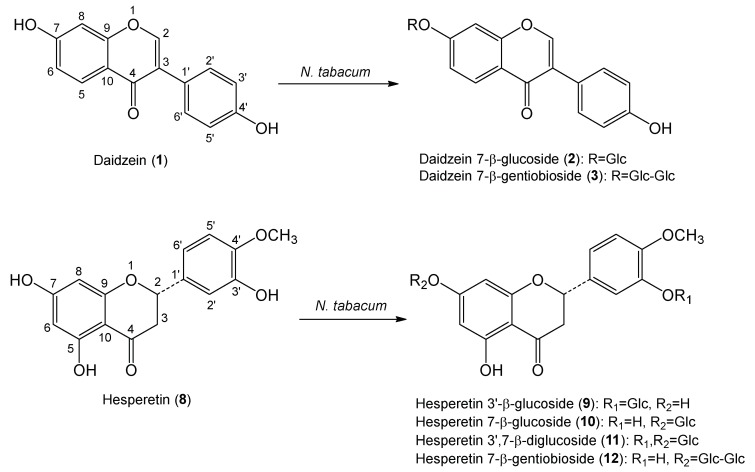
Synthetic schemes of glycosides of daidzein and hesperetin.

**Figure 3 molecules-24-02975-f003:**
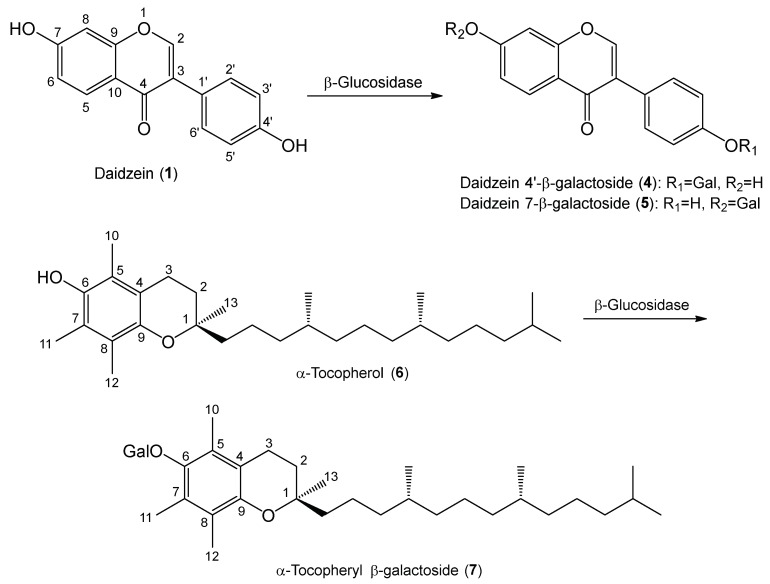
Synthetic schemes of galactosides of daidzein and α-tocopherol.

**Figure 4 molecules-24-02975-f004:**
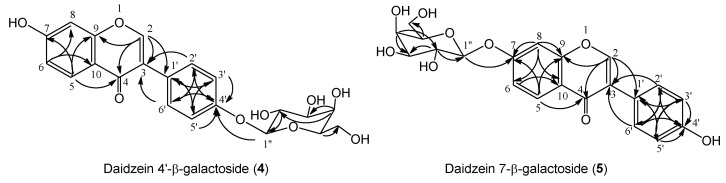
HMBC correlations of daidzein galactosides.

**Table 1 molecules-24-02975-t001:** Anti-allergic activities of compounds **1**–**5**.

Compound	Histamine Release-Inhibiting Activity / %inhibition ^1^
**Kaempferol**	80
**1**	58
**2**	62
**3**	67
**4**	70
**5**	76
**6**	61
**7**	82
**8**	38
**9**	40
**10**	45
**11**	63
**12**	69

^1^ Histamine release-inhibiting activity is expressed as %inhibition.

**Table 2 molecules-24-02975-t002:** Tyrosinase inhibitory activities of compounds **1**–**5**.

Compound	Tyrosinase Inhibitory Activity IC_50_ ^1^ / μM
**Kojic acid**	35 ± 15
**1**	392 ± 88
**2**	303 ± 45
**3**	280 ± 33
**4**	125 ± 41
**5**	102 ± 39
**6**	510 ± 108
**7**	54 ± 25
**8**	437 ± 76
**9**	355 ± 32
**10**	318 ± 27
**11**	176 ± 35
**12**	139 ± 14

^1^ Tyrosinase inhibitory activity is expressed as the 50% inhibitory concentration (IC_50_). The results are shown as the mean ± standard deviation from triplicate experiments.

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
