# Peer review of "Synthesis of Daidzein Glycosides, α-Tocopherol Glycosides, Hesperetin Glycosides by Bioconversion and Their Potential for Anti-Allergic Functional-Foods and Cosmetics"

_molecules, 2019, doi:10.3390/molecules24162975_

Round 1
Reviewer 1 Report
The writing of the text is good enough. The work presented in MS sounds interesting as well. It may be considered for publication after some modifications to the article. The comments are given below.
The reference of known compounds 2-3 and 9-12 as they compared needed to be added.
The authors indicated the rationale that they synthesized glycosylated-diadzein, hesperetin, and α-tocherol to ehance their PK. However, they provided no informationregadrong the test of the activity against histamine release and tyrosinease. More description should be included in the introduction.
The authors should explain that using beta-glucosidase exclusively gave the galactose-conjugated products, but with no gllucosylated ones.
It is suggested that key HMBC correlations of two new compounds 4 and 5 in the Figure or Table.
The reference compounds should be included in the assay of histamine release inhibitory activity in Table 1. Also, it should be added to the Table 2.
The authors will provide the description structure-activity relationship (SAR) of the three series compounds.
In the material and methods section, the coupling "J" should be italic. Additionally, the spectroscopic data of known compounds isolated should be provided.
Reviewer 2 Report
Polyphenols glycosilation by cultured plant cell is a very interesting topic. In the present article the author described the identification of glycosides derivates but not sufficently described the cells cultivation (i.e. cultural medium, substrate). Also introduction can be implemented.
Reviewer 3 Report
Hama and co-workers' manuscript deal with the enhancement of the bioactivity of a series of isolavones, e.g., daidzein, alpha-tocopherol and hesperetin, by its glycosylation. Although the topic might be of interest for the Molecule's audience, this referee feels the paper needs to be significantly improved prior to its publications. My major comments are:
The anti-allergic activities listed in Table 1 and the inhibitory action depicted in Table 2 need to be confirmed by comparing the obtained results for the parent molecules with data available in the literature. Authors need to proof the accuracy of their experimental protocol.
I cannot see any control measure.
Other effects, as pH and temperature, have been not accounted for.
Author need to assess also the effect of isolated GLA (anti-allergic and the inhibitory activities) to demonstrate the synergic rather than the combined effect of metabolites.
The topic is poorly introduced. Indeed, introduction section needs to be further improved. Authors needs to correctly review the previous use of GLA and/or other modification in the parent isolavone compounds.
The main issue of the current paper is the lack of novelty. That procedure has been extensively used, see for instance Curr Org Chem. 2017, 21, 218 and Catalysts 2018, 8, 387. Such papers are here suggested as examples and are not work by this referee. An additional effort by authors is needed to demonstrate that the reported data are expected to impact into the field.
In my opinion, all suggested changes are doable and consequently I recommend reconsider after mayor revision.
Reviewer 4 Report
The manuscript described synthesis of four derivatives of daizein, one α-tocopherol and four of hesperetin. Compounds were tested as antiallergic agents and tyrosinase inhibitors. Authors should give attention to the following remarks point-by-point:
1. Introduction is too short. Please add more information about daizein, α-tocopherol and hesperetin. This section need to contain figure which presented structure of compounds.
2. Scheme of synthesis should be added. Synthesis should be described with more detail.
3. Lines 110-136 should be a separate section. Authors need to added 2D spectra to main text of manuscript.
4. The obtain results should be more precisely discussed.
5. Line 203: EtOAc should be change to AcOEt
6. Conclusion should be change
7. References should contain doi
Author Response
>Introduction is too short. Please add more information about daizein, α-tocopherol and hesperetin.This section need to contain figure which presented structure of compounds.
Now, introduction is not short. In Introduction, more information about substrates and their figure have been added (lines 41,42,47, Figure 1).
>Scheme of synthesis should be added. Synthesis should be described with more detail.
Synthesis schemes have been added (Figure 2). Bioconversion conditions such as cells cultivation have been added (lines 246-251).
>Lines 110-136 should be a separate section. Authors need to added 2D spectra to main text of manuscript.
Sections “2.2.1. Synthesis of daidzein galactosides and a-tocopheryl galactoside” and “2.2.2. Determination of chemical structure of new daidzein galactosides” have been added (lines 100,116). The HMBC correlations of two new glycosides have been added in Figure 2.
>The obtain results should be more precisely discussed.
Discussion has been changed (lines 193-213,220-227,230-233,235-237).
>Line 203: EtOAc should be change to AcOEt
EtOAc has been corrected to AcOEt (please see Section 4.2)
>Conclusion should be change
Conclusion has been changed (lines 306-310,312-315).
>References should contain doi
In Reference, doi has been added.
Furthermore, Title has been changed to “Synthesis of Glycosides of Daidzein, α-Tocopherol, Hesperetin by Bioconversion and their Potential for Anti-allergic Functional-foods and Cosmetics”.

Round 2
Reviewer 1 Report
The revised version is modified accordingly, mostly, based on the referee's comments and suggestions. Despite the weak activity of the isolated cmpounds against histamine release and tyrosinease, the results didn't obstruct the acceptance of the work, in particular the novelty of the two compounds.
Reviewer 3 Report
Although this paper still reports preliminary results, I feel the reached conclusions are now suitable for publication in Molecules with not further major modifications. As a general suggestion, authors should avoid the repetitive/unnecessary use of which were known compounds (Figure 2)". Even if their original intention is to reply one of the referee's comments, such change does not lead to a better understanding of authors' reasoning. The statement "It is not clear if the physiological properties of the compounds are synergic." in lines 236-237 is rather than confusing. It should be clarified or deleted. This referee is also positive that the suggested pH experiments must be performed in the next contribution.
Reviewer 4 Report
The quality of manuscript is better, but I have some comments:
Captains of Fig 1 and Fig 2 should be under figure Fig 2 should be in synthesis part Authors should add figure with correlation in HMBC, like scheme 4 in “The double Smiles rearrangement in neutral conditions leading to oneof 10-(nitropyridinyl)dipyridothiazine isomers” Authors should added explain why they did not check influence of pH and temperature on anti-allergic activities.Author Response
please see the attached file
